# FLoE: Fisher-Based Layer Selection for Efficient Sparse Adaptation of Low-Rank Experts

## Abstract

Parameter-Efficient Fine-Tuning (PEFT) methods have emerged as a widely adopted strategy for adapting pre-trained Large Language Models (LLMs) to downstream tasks, significantly reducing memory and computational costs. However, most existing PEFT techniques uniformly deploy LoRA adapters across all layers, disregarding the intrinsic heterogeneity of layer contributions and task-specific rank requirements. This uniform paradigm leads to redundant parameter allocation and suboptimal adaptation efficiency. To address these limitations, we propose FLoE, a novel PEFT framework that introduces two key innovations: (i) a Fisher information-guided importance scoring mechanism to dynamically identify task-critical transformer layers for MoE-based low-rank adaptation, enabling sparse adapter deployment; and (ii) a Bayesian optimization-driven rank allocator that automatically determines optimal LoRA ranks on specific datasets without exhaustive grid search. Extensive experiments across diverse LLMs and benchmarks reveal that FLoE achieves impressive efficiency-accuracy trade-offs, making FLoE particularly advantageous in resource-constrained environments that necessitate rapid adaptation.

## 1 Introduction

Adapting Large Language Models (LLMs) for multiple downstream tasks traditionally relies on full fine-tuning (FFT), which requires retraining all model parameters. To reduce the training cost, parameter-efficient fine-tuning (PEFT) techniques [35; 13; 7] have been developed, which can be broadly categorized into LoRA-based [13; 7; 29], Adapter-based [48; 24] and Prompt-based [26; 30; 25] approaches. While tuning a limited set of parameters is effective for domain adaptation, PEFT methods like LoRA [15] often exhibit a performance gap compared to the FFT baseline. This gap widens further when tuning on complex datasets [33] with diverse sub-domains and task types, which requires models to distinguish subtle, non-overlapping features while avoiding redundancy.

Recent studies explore a hybrid solution [11; 8; 16], showing that combining LoRA with the Mixture-of-Experts (MoE) [17; 41] is a promising recipe. Among these solutions, HydraLoRA [44] stands out by discovering the asymmetric property of LoRA and implementing $B$ matrices as domain-specific experts, achieving impressive adaptation performance. However, existing methods [44; 52; 29; 8] adopt a uniform placement strategy that indiscriminately deploys fixed-rank LoRA adapters across all transformer layers. Our investigation yields two critical observations that challenge the premise of current implementations:

- Indiscriminate deployment of MoE-based LoRA adapters leads to unnecessary computational overhead as shown in Fig. 1b, revealing a paradoxical trade-off between the number of trainable parameters and overall performance gains.
- LoRA-based tuning is highly sensitive to the choice of rank [52; 19; 46; 32]. Since models trained with one rank do not generalize to others, it is crucial to identify the optimal rank in advance to avoid costly retraining for each possible rank.

Based on these observations, the key to improve MoE-based LoRA tuning is to identify and adapt a small number of critical layers. To quantify the concept of "critical", our motivation is that: if a layer is important for task-specific adaptation, the parameters of its residual trainable adapter should

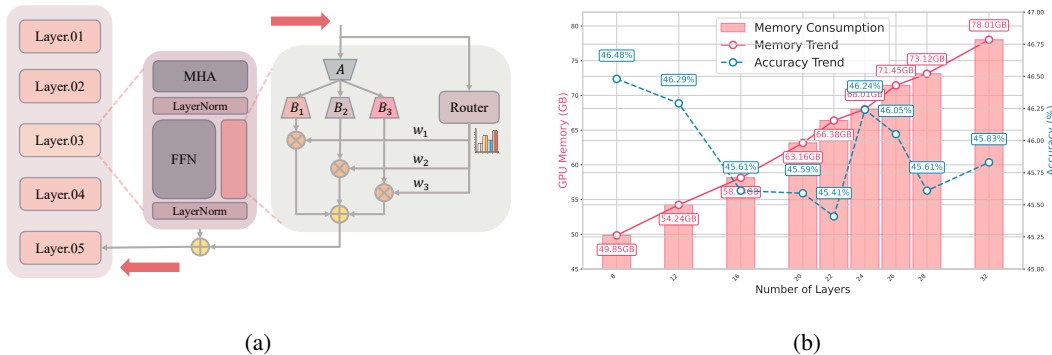

(a)                                                                          (b)

Figure 1: (a) Architecture of our MoE-based LoRA implementation. We adopt the asymmetric architecture of HydraLoRA [44]: a shared $A$ matrix captures general features of the dataset, and multiple distinct low-rank experts $B_i$ learn task-specific patterns. The router takes in an intermediate token representation and generates gating scores $w_i$ over experts. (b) Experimental evidence supporting our motivation: we evaluate GPU memory consumption and adaptation performance of the FLoE layer selection algorithm on LLaMA2-7B fine-tuned with Databricks-Dolly-15K. As the number of adapted layers increases, GPU memory usage grows linearly, yet model accuracy does not improve and even underperforms compared to few-layer adaptation.

exhibit high sensitivity to the adaptation loss, while the corresponding pre-trained weights remain relatively insensitive. From a mathematical view, we can use the variation of gradients to quantify this property, which can be measured with Fisher Information. Another question is how to determine the optimal rank before training. To address these questions, we propose FLoE, a sparse layer adaptation framework that provides a unified selection of layer and LoRA rank. The overall *pipeline* includes: first fine-tune a full-layer model on a sampled dataset using MoE-based LoRA shown in Figure 1a, then applies FLoE to determine critical layers and optimal ranks. During final adaptation on the target dataset, all pre-trained weights are frozen while only the adapters on the critical layers are updated.

Our contributions can be summarized as follow:

- We introduce a Fisher-based importance scoring algorithm that dynamically identifies critical transformer layers for MoE-based low-rank adaptation, enabling sparse, context-aware adapter deployment.

- We incorporate a Bayesian optimization step to estimate the optimal LoRA rank before training on the target dataset, avoiding exhaustive grid search and retraining.

- Experiments show that FLoE achieves comparable or even better performance than prior PEFT methods across diverse datasets and model families, with notable advantages in low-resource and fast-adaptation scenarios. By adapting only 25% of layers, FLoE retains 93.1% of full fine-tuning accuracy on MMLU benchmarks, and achieves a 7.0% relative improvement over the best-performing full-layer methods in mixed-domain adaptation, demonstrating its superior capability in mitigating domain interference while maintaining parameter efficiency.

## 2  RELATED WORK

**Parameter-Efficient Fine-tuning.** Parameter Efficient Fine-Tuning (PEFT) techniques aim to reduce the training costs of the LLMs. Previous PEFT approaches can be broadly classified into the following categories: i) Prefix-tuning [26] and prompt-tuning [25]: prominent approaches that fine-tune continuous prompts rather than discrete ones. ii) Adapter-based tuning: inserts additional adapters into the model or scales activations with learned vectors, including AdaMix [48] and $(IA)^3$ [28]. iii) Low-rank adaptation: introduces trainable low-rank matrices to LLMs, keeping the original weights frozen for efficiency, including LoRA [15] and its variants, such as AdaLoRA [52], HydraLoRA [44] and others [51; 21; 39; 5; 29; 29]. Extensions to multi-LoRA architectures include Multi-Head Routing [37] for Mixture-of-Experts and LoraHub [16] for task composability.

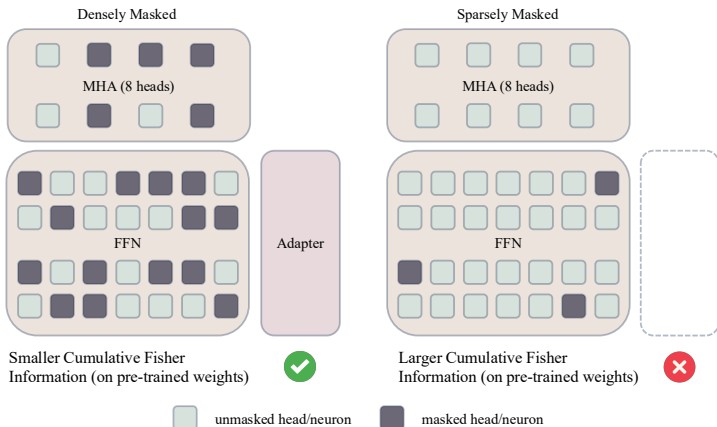

Figure 2: Mechanism of FLoE layer selection. A densely masked layer intends to have higher importance for low-rank adaptation, so we add a residual trainable adapter to the FFN component. The adaptation process only updates the adapter.

**Layer-wise Selective Fine-tuning.** Recent studies [9; 40; 50] have raised the issue of layer redundancy in pre-trained models. Surgical fine-tuning [23] updates only a subset of layers based on domain shift, while SubTuning [20] employs a greedy search to identify the most suitable layers, requiring significant computational resources. LISA [38] bridges the gap between full-parameter tuning and LoRA, introduces a layer-wise importance sampling mechanism during training.

## 3 METHODOLOGY

Let $\mathcal{F}$ denote a pre-trained $L$-layer Transformer model. Given a dataset $\mathcal{D} = \{(x_i, y_i)\}_{i=1}^{N}$, where $x_i$ denotes the input data and $y_i$ the corresponding label. For layer $k \in \{1, \ldots, L\}$, let $\theta_k$ denote its pre-trained weights. LoRA introduces trainable low-rank matrices $A_k \in \mathbb{R}^{d \times r}$ and $B_k \in \mathbb{R}^{r \times d}$ to approximate weight updates $\phi_k = B_k A_k$. Here we follow HydraLoRA [44] to use an MoE-based architecture, extending LoRA by employing a shared $A$ matrix and $M$ parallel low-rank experts $\{B_k^{(i)}\}_{i=1}^{M}$ alongside a router network (implemented as a dense layer followed by a softmax function). Suppose the router outputs a vector of contribution weights $\{\omega_k^{(i)}\}_{i=1}^{M}$ based on the intermediate token representation. The weight updates are formulated as:

$$\phi_k = \sum_{i=1}^{M} \omega_k^{(i)} B_k^{(i)} A_k \tag{1}$$

The final merged weights are $W_k = \theta_k + \phi_k$.

### 3.1 PROBLEM FORMULATION

As our goal is to find a subset of layers $\mathcal{S} \subseteq \{1, 2, \ldots, L\}$ to add trainable adapters, we quantify the contribution of each layer to the model adaptation performance via an importance score $s_k$. To achieve this, we introduce a binary mask variable $\mathbf{m}_k \in \{0, 1\}^{|\theta_k|}$ as an intermediate to calculate $s_k$ during adaptation. The adapted weight $\tilde{\theta}_k$ is then computed as:

$$\tilde{\theta}_k = \mathbf{m}_k \odot \theta_k + \phi_k, \tag{2}$$

As shown in Figure 2, the mask variable $\mathbf{m}_k$ is applied to the pre-trained weights $\theta_k$, based on the principle that mask sparsity reflects the necessity of adaptation. If a pre-trained weight $\theta_{k,i}$ is masked ($\mathbf{m}_{k,i} = 0$), the adapter weights must compensate for its removal to maintain performance. If $\theta_{k,i}$ is retained ($\mathbf{m}_{k,i} = 1$), then $\phi_{k,i}$ only serves as a residual correction. Thus, the sparsity of $\mathbf{m}_k$ directly correlates with the contribution of $\phi_k$. Let $Z(\mathbf{m}_k)$ denote the number of masked weights. Higher sparsity ($Z(\mathbf{m}_k) \ll Z(\mathbf{m}_k)$) indicates that $\theta_k$ are poorly aligned with the dataset,

requiring significant adapter intervention. Lower sparsity $(Z(\mathbf{m}_k) \ll Z(\mathbf{m}_k))$ suggests the pre-trained knowledge in layer $k$ remains largely valid, requiring minimal adaptation. This explicitly disentangles pre-trained knowledge retention from task-driven adaptation.

Let $\boldsymbol{\theta} \triangleq [\theta_1, \ldots, \theta_L]$ and $\boldsymbol{\phi} \triangleq [\phi_1, \ldots, \phi_L]$. The fine-tuning process only updates $\boldsymbol{\phi}$ while keeping $\boldsymbol{\theta}$ frozen. After obtaining $\boldsymbol{\theta}$ and $\boldsymbol{\phi}$, we optimize the mask variable $\mathbf{m}$ to find the optimal $\mathcal{S}$ under the following constrained objective:

$$\arg\min_{\mathbf{m}} \mathcal{L}(\mathbf{m}; \boldsymbol{\theta} + \boldsymbol{\phi}) \quad \text{s.t.} \quad \text{Cost}(\mathbf{m}; \boldsymbol{\theta}) \leq C \tag{3}$$

## 3.2 FISHER INFORMATION-AWARE ESTIMATION

To enable gradient-based optimization for the constrained problem Eq. 3, the cost function should be differentiable with respect to the mask $\mathbf{m}$. Here, we use Taylor importance [34] as the cost function, which measures the sensitivity of the next-token prediction (NTP) loss to parameter perturbations. This allows us to identify parameters that have minimal influence on the base model prediction, as indicated by the deviation in the next-token prediction loss. For layer $k$, its element-wise Taylor Importance $T_k$ is defined as:

$$T_k = \sum_i \left| \frac{\partial \mathcal{L}_{\text{NTP}}}{\partial \theta_{k,i}} \theta_{k,i} \right|, \tag{4}$$

For simplicity we deonte the cost function as $\text{Cost}(\mathbf{m})$. By constraining the total cost $\sum_{k \in \mathcal{S}} T_k$, we prioritize adapting layers with lower $T_k$ (i.e., those less critical to the pre-trained knowledge). This ensures that adaptation focuses on "safe" regions of the network, reducing the risk of overwriting crucial pre-trained features. Therefore, the cost function can be formulated as:

$$\text{Cost}(\mathbf{m}) = \sum_{k \in \mathcal{S}} \sum_i T_{k,i} \cdot \mathbf{m}_{k,i}. \tag{5}$$

**Taylor Approximation of the Task-Specific Loss Function.** We start by analyzing the sensitivity of the loss function $\mathcal{L}$ to the mask variable $\mathbf{m}$. Assuming we have local smoothness around $\mathbf{m} = \mathbb{1}$, then the loss can be approximated using a second-order Taylor expansion:

$$\mathcal{L}(\mathbf{m}; \tilde{\boldsymbol{\theta}}) \approx \mathcal{L}(\mathbb{1}; \tilde{\boldsymbol{\theta}}) + \frac{1}{2}(\mathbb{1} - \mathbf{m})^{\top} \mathbf{H}(\mathbb{1} - \mathbf{m}), \tag{6}$$

where $\tilde{\boldsymbol{\theta}} = \boldsymbol{\theta} + \boldsymbol{\phi}$ denotes the merged weights. Here, the first-order term $\nabla \mathcal{L}(\mathbb{1}; \tilde{\boldsymbol{\theta}})^{\top}(\mathbf{m} - \mathbb{1}) = 0$ due to the assumption that the model has converged to a local minima, where the gradient term is close to 0 [10; 47; 22]. As $\mathcal{L}(\mathbb{1}; \tilde{\boldsymbol{\theta}})$ is a constant, we can rewrite the optimization objective in Eq. 6 as follows:

$$\arg\min_{\mathbf{m}} \mathcal{L}(\mathbf{m}) \approx \arg\min_{\mathbf{m}} (\mathbb{1} - \mathbf{m})^{\top} \mathbf{H}(\mathbb{1} - \mathbf{m}). \tag{7}$$

Eq. 7 shows that the optimal mask is determined by the Hessian of the loss with respect to the mask variables, i.e. $\mathbf{H} = \mathbb{E}_{x \sim \mathcal{D}}[\nabla^2_{\mathbf{m}} \mathcal{L}(\mathbb{1}; \tilde{\boldsymbol{\theta}})]$. Since computing the exact Hessian matrix is infeasible, we approximate the Hessian $\mathbf{H}$ with the empirical Fisher Information Matrix (FIM), which is defined as:

$$\mathcal{I}(\mathbf{m}) = \mathbb{E}_{x \sim \mathcal{D}} \left[ \nabla_{\mathbf{m}} \mathcal{L}(\mathbb{1}) \nabla_{\mathbf{m}} \mathcal{L}(\mathbb{1})^{\top} \right]. \tag{8}$$

**Diagonal Approximation of the FIM.** Assuming each layer $k$ contains $|\theta_k|$ parameters (including the weight parameters of both MHA and FFN components), then $\mathbf{m}_k$ can be seen as a vector of length $|\theta_k|$. As $\mathbf{m}$ is applied across all $L$ layers, the full FIM $\mathcal{I}$ has $L^2 |\theta_k|^2$ elements, making its computation and storage intractable for large values of $|\theta_k|$. To address this challenge, we adopt a diagonal approximation of $\mathcal{I}$, reducing its complexity from $O(L^2 |\theta_k|^2)$ to $O(L |\theta_k|)$. This approximation is based on an assumption that cross-layer interactions can be neglected, since the off-diagonal terms $\mathcal{I}_{k,l}$ $(k \neq l)$ are ignored. Under this assumption, only the diagonal elements $\mathcal{I}_{k,k}$ are computed for each layer $k$, where:

$$\mathcal{I}_{k,k} = \mathbb{E}_{x \sim \mathcal{D}} \left[ \frac{\partial \mathcal{L}}{\partial \mathbf{m}_k} \right]^2. \tag{9}$$

This further simplifies Eq. 7 as follows:

$$\arg\min_{\mathbf{m}} \mathcal{L}(\mathbf{m}) \approx \arg\min_{\mathbf{m}} \sum_{k=1}^{L} (\mathbb{1} - \mathbf{m}_k)^2 \mathcal{I}_{k,k}$$

$$= \arg\min_{\mathbf{m}} \sum_{k=1}^{L} \sum_{i=1}^{|\theta_k|} (1 - \mathbf{m}_{k,i})^2 \mathcal{I}_{k,k,i}. \tag{10}$$

Let $Z_k(\mathbf{m}) = \{i : \mathbf{m}_{k,i} = 0\}$. Since we restrict the possible mask values to either 0 or 1, the following can be derived from Eq. 10:

$$\arg\min_{\mathbf{m}} \mathcal{L}(\mathbf{m}) \approx \arg\min_{\mathbf{m}} \sum_{k} \sum_{i \in Z_k(\mathbf{m})} \mathcal{I}_{k,k,i}. \tag{11}$$

Then the optimization objective in Eq. 3 is equivalent to minimizing the sum of layer-wise Fisher information of the masked parameters:

$$\arg\min_{\mathbf{m}} \sum_{k} \sum_{i \in Z_k(\mathbf{m})} \mathcal{I}_{k,k,i} \quad \text{s.t.} \quad \sum_{k} \sum_{i \in Z_k(\mathbf{m})} T_{k,i} \leq C. \tag{12}$$

### 3.3 SOLVING THE CONSTRAINED OPTIMIZATION PROBLEM

**Determination Stage.** Within each transformer layer, the architecture consists of two primary components: a multi-head attention (MHA) module and a feed-forward network (FFN). We denote the mask variables for these components in layer $k$ as $\mathbf{m}_k^{\text{MHA}}$ and $\mathbf{m}_k^{\text{FFN}}$, respectively. Suppose there are $N^{\text{MHA}}$ head mask variables and $N^{\text{FFN}}$ neuron mask variables.

The optimization problem in Eq. 12 can be interpreted as follows: For each layer $k$, we seek a subset of unmasked parameters $\mathcal{M}_k$ that minimizes their total Fisher information on the adapted model $\mathcal{F}(\tilde{\boldsymbol{\theta}})$), while constraining their total Taylor importance (computed on the base model $\mathcal{F}(\boldsymbol{\theta})$) under a global budget $C$.

As the Fisher information and Taylor importance vary across individual parameters, Eq. 12 becomes a dynamic programming problem which is memory inefficient. To reduce the 2-dimensional search space into a linear form, we employ a component-wise approximation within the same layer. Specifically, we compute the scores for MHA and FFN respectively, and then average these values to yield a single importance estimate per parameter. This allows us to use a greedy solution (described in Algorithm. 1). The algorithm iteratively excludes the parameters with smallest Taylor importance until the budget $C$ is reached, while maximizing the cumulative Fisher information of the included parameters.

**Refinement Stage.** The component-wise approximation in determination stage decouples the selection of MHA heads and FFN neurons within each layer, thereby ignoring potential interactions between them. While efficient, this approximation may lead to suboptimal trade-offs between Taylor importance and Fisher information. To mitigate this, we propose a post-hoc refinement stage that jointly optimizes the masks for both components under the same global budget constraint. This refinement operates on the initial greedy solution as a warm start, enabling recovery of near-optimal masks with minimal computational overhead.

The refinement process is designed to iteratively adjust the selected MHA heads and FFN neurons while respecting the budget constraint. Let $\mathcal{M}_h^*$ and $\mathcal{M}_f^*$ denote the sets of unmasked MHA heads and FFN neurons from the greedy solution. We define the refinement loss for a candidate mask pair $(\mathcal{M}_h, \mathcal{M}_f)$ as:

$$\mathcal{L}(\mathcal{M}_h, \mathcal{M}_f) = \sum_{i \notin \mathcal{M}_h} \tau_i^h + \sum_{j \notin \mathcal{M}_f} \tau_j^f, \quad \text{s.t.} \quad \sum_{i \notin \mathcal{M}_h} t_h + \sum_{j \notin \mathcal{M}_f} t_f \leq C. \tag{13}$$

The goal is to perturb $(\mathcal{M}_h^*, \mathcal{M}_f^*)$ to minimize $\mathcal{L}$ under the constraint. Let $\mathcal{C}$ denote a joint candidate set containing all parameters (both masked and unmasked) at the same layer:

$$\mathcal{C} = \left\{ (i,j) \mid i \in \mathcal{M}_h^*, j \in \mathcal{Q}_{\text{FFN}} \setminus \mathcal{M}_f^* \right\} \cup \left\{ (i,j) \mid i \in \mathcal{M}_f^*, j \in \mathcal{Q}_{\text{MHA}} \setminus \mathcal{M}_h^* \right\}. \tag{14}$$

---

**Algorithm 1** Greedy Mask Search

---

1: **Input:** Budget $C$, Fisher score per-head $\{\tau_i^h\}_{i=1}^{N^{\text{MHA}}}$, Fisher score per-neuron (for FFN) $\{\tau_j^f\}_{j=1}^{N^{\text{FFN}}}$, Taylor score per-head $t_h$, Taylor score per-neuron $t_f$.

2: Initialize optimal Fisher loss $\mathcal{L}^* \leftarrow \infty$, optimal sets of unmasked indices $(\mathcal{M}_h^*, \mathcal{M}_f^*) \leftarrow (\emptyset, \emptyset)$, mask variables $(\mathbf{m}^{\text{MHA}}, \mathbf{m}^{\text{FFN}}) \leftarrow (\mathbb{1}, \mathbb{1})$

3: **for** $n = 0$ to $N^{\text{MHA}}$ **do**

4:     Compute cost for MHA module: $\hat{C}_h = n \cdot t_h$

5:     **if** $\hat{C}_h > C$ **then**

6:         **continue**                                               ▷ Exceeds budget

7:     **end if**

8:     Remaining budget: $C_r = C - \hat{C}_h$

9:     Retained neurons: $f = \min\left(\max\left(0, \left\lfloor \dfrac{C_r}{t_f} \right\rfloor\right), N^{\text{FFN}}\right)$

10:    Select $n$ heads with smallest $\tau_i^h$: indices $\mathcal{P}_h$

11:    Select $f$ neurons with smallest $\tau_j^f$: indices $\mathcal{P}_f$

12:    Compute total loss: $\mathcal{L} = \sum_{i \in \mathcal{P}_h} \tau_i^h + \sum_{j \in \mathcal{P}_f} \tau_j^f$

13:    **if** $\mathcal{L} < \mathcal{L}^*$ **then**

14:       $\mathcal{L}^* \leftarrow \mathcal{L}, (\mathcal{M}_h^*, \mathcal{M}_f^*) \leftarrow (\mathcal{P}_h, \mathcal{P}_f)$

15:    **end if**

16: **end for**

17: Apply masks: Set $\mathbf{m}^{\text{MHA}}[\mathcal{M}_h^*] = 0$, $\mathbf{m}^{\text{FFN}}[\mathcal{M}_f^*] = 0$

18: **Output:** Optimal mask $\mathbf{m}^* = (\mathbf{m}^{\text{MHA}}, \mathbf{m}^{\text{FFN}})$

---

where $\mathcal{Q}_{\text{MHA}}$ and $\mathcal{Q}_{\text{FFN}}$ represent the complete sets of parameters in MHA and FFN modules, respectively. For each candidate parameter $p \in \mathcal{C}$, compute the *swap gain* if $p$ is masked and another parameter $q$ (of any component) is unmasked to compensate for the budget:

$$\Delta\mathcal{L}_{p \to q} = \tau_p - \tau_q, \quad \Delta C_{p \to q} = t_p - t_q. \tag{15}$$

A valid swap satisfies $\Delta C_{p \to q} \geq 0$ to preserve the budget constraint. Then we select the swap with the largest $\Delta\mathcal{L}_{p \to q}$ (i.e., maximal reduction in total Fisher loss) to update $\mathcal{M}_h^*$, $\mathcal{M}_f^*$ and the remaining budget. Repeat this process until no improving swaps exist. The algorithm is implemented in Algorithm. 2.

The refinement stage approximates a single iteration of the Lagrange multiplier method, where swaps implicitly adjust the balance between Taylor importance (constraint) and Fisher information (objective). By restricting swaps to the vicinity of the initial greedy solution, it avoids the $O(|\tilde{\boldsymbol{\theta}}||\boldsymbol{\theta}|)$ complexity of full dynamic programming while recovering Pareto-improved solutions.

**Tuning Stage.** Since our goal is to use mask values to measure parameter importance, the initial binary masks are insufficient, as they fail to capture parameters that contribute marginally on their own but are collectively important. To address this limitation, we introduce a differentiable tuning stage that relaxes the binary masks into continuous values. The layer-wise reconstruction objective is formulated as the residual activation difference:

$$\underset{\mathbf{m}_k}{\arg\min} \|\mathcal{O}(x; \mathbb{1}) - \mathcal{O}(x'; \mathbf{m}_k)\|_2^2 \tag{16}$$

where $x'$ and $x$ are inputs to the layer with or without mask, $\mathcal{O}(x; \mathbf{m}_k) = x + l_k(x; \mathbf{m}_k)$ denotes the residual output of a mask-scaled layer, and $l_k$ indicates a MHA or FFN layer. A detailed derivation is provided in Appendix H.

## 3.4 DYNAMIC RANK SELECTION FOR LORA-BASED PEFT

We employ Bayesian optimization primarily to determine the optimal LoRA rank $r$ across layers, while also adjusting the number of experts in MoE-LoRA as a secondary objective. Bayesian optimization is particularly advantageous in multi-dimensional hyperparameter spaces, as it constructs a probabilistic surrogate model to approximate the relationship between hyperparameters and validation

performance. Using an acquisition function, BO iteratively proposes new candidate configurations by leveraging past evaluation results, thereby focusing computational resources on regions most likely to yield improved performance. Compared to grid or random search, BO significantly reduces the number of expensive fine-tuning trials and enables early elimination of suboptimal configurations. This makes it especially suitable for tuning costly PEFT setups with multiple hyperparameters. Table 8 highlights the efficiency gains of BO when tuning LoRA rank $r \in [4, 64]$ (for rank allocation) and the number of experts $\in [2, 10]$.

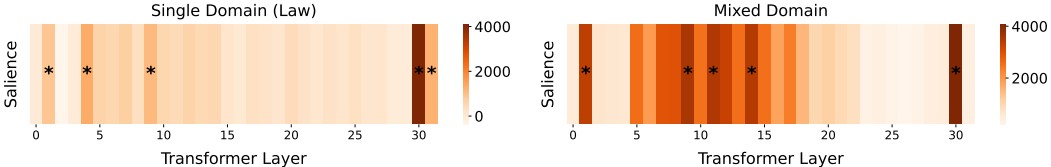

Figure 3: Saliency maps illustrating the importance of layers on the single-domain (Lawyer-Instruct) and mixed-domain/task (FLANv2) datasets. The base model is LLaMA2-7B. Black dots indicates the top-5 critical layers for the given task.

Table 1: Performance comparison of PEFT methods on LLaMA2-7B. Performance metrics include the accuracy on MMLU (5-shot) and GSM8K, and Pass@1/Pass@10 on HumanEval. LoRA and its variants (AdaLoRA, DoRA and HydraLoRA) adopt a single A matrix, but differ in the rank and the number of B matrices, which are determined by Bayesian optimization.

| Schemes | MMLU | Medical | Law | HumanEval | | GSM8K | Avg. Rank | Avg. #B |
| --- | --- | --- | --- | --- | --- | --- | --- | --- |
| | | | | P@1 | P@10 | | | |
| LLaMA2-7B [45] | 38.79 | 36.05 | 33.64 | 13.17 | 20.41 | 10.44 | - | - |
| Full Fine-Tuning | 49.91 | 46.76 | 46.22 | 20.24 | 32.93 | 25.69 | - | - |
| Prompt Tuning [25] | 39.97 | 37.46 | 34.88 | 13.59 | 21.62 | 13.25 | - | - |
| P-Tuning$_{(256)}$ [31] | 41.02 | 39.85 | 36.64 | 13.53 | 21.20 | 15.50 | - | - |
| Prefix Tuning [26] | 41.86 | 40.28 | 36.30 | 13.15 | 22.48 | 16.83 | - | - |
| LoRA [15] | 45.88 | 46.76 | 37.16 | 14.57 | 29.88 | 18.24 | 16 | 1 |
| AdaLoRA [52] | 44.26 | 42.39 | 39.36 | 14.74 | 23.85 | 19.44 | $12 \rightarrow 4$ | 1 |
| DoRA [29] | 44.57 | 44.23 | 38.74 | 14.65 | 24.20 | 19.50 | 10 | 1 |
| HydraLoRA [44] | 45.83 | 46.90 | 37.76 | 14.39 | 28.66 | 19.66 | 8 | 4 |
| **Ours** | **46.48** | **49.15** | **39.14** | **14.82** | **31.71** | **20.09** | 8 | 4 |

## 4 EXPERIMENTS

### 4.1 EXPERIMENTAL SETUP

**Datasets and Benchmarks.** To investigate the effectiveness of our layer selection policy, we conduct experiments on both single- and multi-domain datasets. **Single domain** includes: *General*, *Medical*, *Legal*, *Code Generation*, *Mathematics*. **Multi domain** includes *FLANv2* and we evaluate the performance on BBH benchmark [42]. Detailed descriptions of the datasets and benchmarks are provided in Appendix D.

**Baselines.** To evaluate the adaptation performance on FLoE-selected layers, we compare it with different PEFT methods: *Full Fine-Tuning*, *Prompt Tuning* [25], *P-Tuning* [31], *LoRA* [15], *AdaLoRA* [52], *DoRA* [29], *HydraLoRA* [44]. To evaluate the layer selection policy, we compare FLoE with *Random Selection* and *LISA* [38] (fine-tuned with MoE-LoRA). We further compare FLoE with two LoRA derivatives, *LoraHub* [16] and *LoRAMoE* [8], which also utilize a routing mechanism to coordinate multiple LoRA experts. Detailed descriptions of these baseline methods are provided in Appendix D.2.

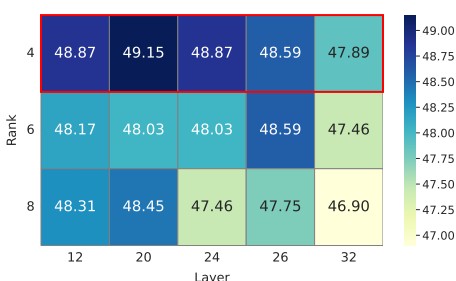
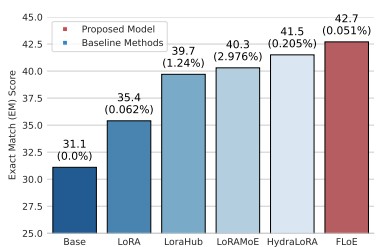

Figure 4: Dynamic rank selection results for the medical task on LLaMA2-7B. The optimal rank ($r = 4$) is determined using Bayesian optimization on a full-layer adapted model (1A/4B).

Figure 5: Mixed domain results evaluated on the BBH benchmark (3-shot) using LLaMA2-7B. FLoE achieves the highest Exact Match (EM) score with the lowest training parameter size.

Table 2: Performance comparison of different layer selection policies, including random selection (with seed 42), LISA + MoE-LoRA and our proposed FLoE training for 1 epoch with LLaMA2-7B.

|  | Random | LISA + MoE-LoRA | Ours (FLoE) |
|---|---|---|---|
| **General** | 46.15 | 46.39 | **46.48** |
| **Medical** | 47.46 | 48.17 | **49.15** |

## 4.2 MAIN RESULTS

**Implementation Details.** For single-domain adaptation, we first fine-tune a fully adapted model on 50% of the target dataset (i.e. the warm-up phase). The LoRA rank and the number of B matrices are optimized via Bayesian optimization, where the rank is searched within $[2, 32]$ (step size 2), and the number of B matrices within $[2, 4]$ (step size 1). We then apply the layer selection algorithm to identify critical layers, followed by fine-tuning a sparsely adapted model on the full target dataset. For mixed-domain adaptation, we sample 1.25% of the FLANv2 dataset for the warm-up phase. Bayesian optimization is applied with the same search range for the rank, while the number of B matrices is searched within $[6, 10]$ (step size 2). The model is then fine-tuned on a larger subset of FLANv2 comprising 3% of the data.

**Results on Single/Mixed Domain dataset.** The experimental results are presented in Table 1 for fine-tuning performance evaluation and Table 2 for layer selection policy evaluation. These results demonstrate that FLoE consistently outperforms all competing approaches while reducing a large portion of trainable parameters. Results in Table 3 show that the optimal number of layers varies depending on the specific domain, with fewer layers generally performing better on MMLU and Medical benchmarks, while a moderate number of layers might be more effective for Law and GSM8K benchmarks. Figure 5 shows the results of mixed tasks and domains. The A/B configuration is 48/48 for LoraHub and LoRAMoE, 1/1 for LoRA, 1/6 for FLoE and HydraLoRA. Detailed results for each task in BBH evaluation are provided in the Appendix 9. The visualizations of the layer importance on single and mixed domain dataset is represented in Figure 3.

**Hyperparameter Tuning via Bayesian Optimization.** We employ Bayesian optimization via Optuna [1], searching in 30 trials for the combination of $r$ and $N_B$. We implement with the Tree-structured Parzen Estimator (TPE) surrogate model and the Expected Improvement (EI) acquisition function. Table 8 shows the efficiency of using Bayesian optimization compared with random search and grid search over a substantially larger search space. Figure 4 compares the performance of fine-tuning different subsets of layers under different ranks ($r = 4, 6, 8$). Here we fix the number of B matrices to 4, since the search space for $N_B$ is very small and the optimal value consistently remains 4 when the validation loss reaches its minimum at $r = 4, 6, 8$. We visualize the convergence of Bayesian optimization process in Figure 8.

**Results on Other Models Families.** To validate the generalization of FLoE, we extend experiments to Gemma2-2B [43], Mistral-7B [18] and LLaMA3.1-8B [12]. As shown in Table 4, both models achieve optimal task performance when adapting 8 layers. Figure 10 visualizes layer importance

Table 3: Layer-specific fine-tuning using FLoE on LLaMA2-7B. We apply a 1A/4B adapter for each selected layer. Best results per column are bolded.

| Layers | MMLU | Medical | Law | HumanEval | | GSM8K | % Param |
|--------|------|---------|-----|-----------|-----------|-------|---------|
| | | | | P@1 | P@10 | | |
| 32 | 45.83 | 46.90 | 37.76 | 14.39 | 28.66 | 17.66 | 0.124 |
| 26 | 46.05 | 47.75 | 38.07 | 14.45 | 27.44 | 19.94 | 0.101 |
| 24 | 46.24 | 47.46 | 37.76 | 14.76 | **31.71** | **20.09** | 0.093 |
| 20 | 45.59 | 47.75 | **39.14** | **14.82** | 31.10 | 18.62 | 0.078 |
| 16 | 45.61 | 48.03 | 38.00 | 13.90 | **31.71** | 18.65 | 0.062 |
| 12 | 46.29 | 48.45 | 37.89 | 13.78 | 29.27 | 19.11 | 0.047 |
| 8 | **46.48** | **49.15** | 36.80 | 13.48 | 29.88 | 17.91 | 0.031 |

Table 4: End-to-end inference latency by running the full MMLU benchmark (including 14,042 examples) on the fine-tuned Gemma2-2B and Mistral-7B models, the configuration for multi LoRA head is 1A/4B. Models are fine-tuned with Dolly-15K. Layer selection is performed using FLoE.

| Models | Metrics | Single LoRA Head | Multi LoRA Heads | | | | | | |
|--------|---------|------------------|-----|-----|-----|-----|-----|-----|-----|
| | | | 32 | 28 | 24 | 20 | 16 | 12 | 8 |
| Gemma2-2B | % Performance (Acc) | 51.25 | - | 51.56 | 51.73 | 51.67 | 51.12 | 51.30 | $51.99^{\uparrow 0.74}$ |
| | Inference Lat. (s) | **1717.90** | - | 3832.37 | 3670.04 | 3368.81 | 3123.01 | 2777.69 | $2516.51^{\uparrow 798.61}$ |
| Mistral-7B | % Performance (Acc) | 60.95 | 53.18 | 61.42 | 61.04 | 60.40 | 61.10 | 61.52 | $62.14^{\uparrow 1.19}$ |
| | Inference Lat. (s) | **1462.27** | 4022.30 | 2055.37 | 1917.85 | 1758.16 | 1666.58 | 1636.02 | $1519.14^{\uparrow 56.87}$ |
| LLaMA3.1-8B | % Performance (Acc) | 62.03 | 61.54 | $62.46^{\uparrow 0.43}$ | 62.36 | 62.25 | 62.45 | 61.88 | 62.02 |
| | Inference Lat. (s) | **1333.49** | 1900.04 | $1796.27^{\uparrow 462.78}$ | 1706.96 | 1611.51 | 1525.08 | 1432.32 | 1145.19 |

Table 5: Training runtime for full-layer and selected-layer adaptation. The FLoE(Total) is the sum of Selected-Layer Adaptation and FLoE Layer Selection.

| | Dolly-15K | Clinic-10K | Lawyer-Instruct | CodeAlpaca |
|--|-----------|------------|-----------------|------------|
| Full-Layer Adaptation (Baseline) | 20105.08 | 20105.08 | 10137.51 | 45092.51 |
| Selected-Layer Adaptation | 16455.04 | 6697.16 | 6895.77 | 16197.23 |
| FLoE Layer Selection | 122.86 | 125.45 | 123.13 | 127.39 |
| FLoE (Total) | **16577.90** (-3527.18) | **6822.61** (-7281.19) | **7018.90** (-3118.61) | **16324.62** (-28767.89) |

distributions, showing that FLoE identifies middle layers for Mistral-7B and deep layers for Gemma2-2B as critical for adaptation.

**Inference Latency.** As MoE-LoRA inherently has more parameters than vanilla LoRA, we compare the latency of running inference on full-layer adapted models with single LoRA head (per adapter) and selected-layer adapted models with multiple LoRA heads (per adapter). Experiments are running with a batch size of 16. While the full-layer LoRA-adapted models achieve minimal inference time, our FLoE layer selection strategy significantly optimizes latency for multi-head configurations. As shown in Table 4, FLoE reduces inference latency by strategically limiting the number of adapted layers.

**Trade-off bewteen Training and Layer Selection.** Table 5 shows the runtime of FLoE layer selection process. The results show that the overall runtime including FLoE layer selection process and selected-layer adaptation is still lower than full-layer adaptation.

## 5 CONCLUSION

In this work, we first discuss the limitations of deploying MoE-based LoRA modules on all transformer layers indiscriminately, where domain interference significantly degrades performance across diverse tasks. To address this, we propose FLoE, a novel PEFT method that introduces two key innovations: Fisher information-aware layer selection and Bayesian optimization-driven dynamic rank allocation. Our experiments demonstrate that FLoE offers a scalable and resource-efficient way for adapting LLMs to specialized domains, advancing the practical deployment of LLMs under constrained computational budgets.

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

## A  LLM Usage Statement

In this work, LLMs are not involved in any core aspects of the research, including data collection, experimental design, model development, result analysis, or conclusion formulation. LLMs are only used for language polishing. Their role are limited to improving textual clarity, enhancing logical coherence, refining the precision of academic terminology, and increasing the readability of experimental results. All text refined with the assistance of LLMs was carefully reviewed by the authors to ensure full alignment with the original research intentions and to maintain academic integrity, without introducing inaccuracies or misleading content.

## B  Reproducibility Statement

We provide code in the supplementary material and implementation details in the appendix, covering batch size, learning rate schedules, and optimizer configurations. The models are publicly available, and all external datasets used in our work are either publicly released or cited with appropriate references.

## C  Limitation

While our proposed FLoE algorithm effectively identifies critical layers for MoE-based LoRA fine-tuing, its current implementation limits each LoRA module with a fixed number of low-rank experts. Future work can explore dynamic expert allocation mechanisms, where both the selection of critical layers and the number of experts per layer are jointly optimized, enabling more granular control over model capacity allocation and better computational resource utilization.

## D  Datasets and Baselines

### D.1  Datasets

**Single domain** includes:

- *General*: we fine-tune with Databricks-Dolly-15K dataset [36] for general knowledge mastering and evaluate with all tasks in MMLU [14].
- *Medical*: we fine-tune with GenMedGPT and Clinic-10K [27] for medical applications and evaluate with 3 medical tasks in MMLU, including *clinical knowledge*, *professional medicine* and *college medicine*.
- *Legal*: we fine-tune with Lawyer-Instruct [2] for legal applications and evaluate with 3 legal tasks in MMLU, including *jurisprudence*, *international law* and *professional law*.
- *Code Generation*: we fine-tuned with CodeAlpaca [3] and evaluate with HumanEval [4].
- *Mathematics*: we fine-tune with the training split of GSM8K [6] for mathematical reasoning and evaluate with the test split.

**Multi domain** includes:

- *FLANv2*: we construct the training dataset by sampling equal-proportion subsets from each of the 46 tasks in FLANv2 [49] and evaluate on BBH benchmark [42].

### D.2  Baselines

Baselines for **PEFT**:

- *Full Fine-Tuning*: the default adaptation strategy involves initializing the model with pre-trained weights and updating all parameters via gradient descent. The number of trainable parameters equals the number of pretrained parameters.
- *Prompt Tuning* [25]: adds manually-designed task-specific prompts to the input. The fine-tuning process only updates the prompt parameters while keeping the pre-trained parameters frozen.
- *P-Tuning* [31]: a prompt adds learnable prompt tokens to the input, optimized by a prompt encoder to find a better prompt. The prompt tokens can be added anywhere in the input sequence.

- *LoRA* [15]: decomposes weight updates into low-rank matrices, enabling efficient adaptation with significantly fewer trainable parameters while preserving model performance.
- *AdaLoRA* [52]: dynamically allocates trainable parameters across weight matrices and layers, prioritizing important components instead of uniformly distributing resources as in LoRA.
- *DoRA* [29]: decomposes pre-trained weights into magnitude and direction components, and apply LoRA on the direction component.
- *HydraLoRA* [44]: incorporating asymmetric LoRA adapters across all layers.

Baselines for **layer selection policy**:

- *Random Selection*: layers are chosen uniformly at random during training (with a fixed random seed 42), serving as a naive baseline to assess the necessity of structured selection.
- *LISA + MoE-LoRA* [38]: LISA freezes most intermediate layers and selectively updates only the embedding layer, the language modeling head layer, and a small number of randomly sampled intermediate layers in each optimization step. We update the weights of MoE-LoRA modules on these layers instead of updating the pre-trained weights. The rank and the number of experts are determined by Bayesian optimization.

Baselines for **multi-LoRA architecture**:

- *LoRAMoE*[8]: combines lightweight experts (LoRA) with MoE architecture for high efficiency, generalizing to new tasks without prior knowledge.
- *LoraHub*[16]: employs black-box optimization to learn weights of 20 randomly selected LoRAs for new tasks, using weighted averaging without needing gradient calculations.

## E  HYPERPARAMETER SETTINGS

Table 6: Hyperparameter settings for our experiments and LoRA-based baseline methods. MSL indicates the max sequence length, BSZ indicates the batch size. All methods use the AdamW optimizer. The weight decay is set to 0.

|  | $\alpha$ | Dropout | MSL | Warmup Ratio | BSZ | Epochs | LR | Seed | Where |
|---|---|---|---|---|---|---|---|---|---|
| **Hyperparameter** | 32 | 0.05 | 1024 | 0.03 | 1 | 1 | 2e-4 | 41 | Up,Down,Gate |

## F  ABLATION STUDY

The FLoE layer selection algorithm comprises three steps: (i) determination stage for coarse-grained binary mask value initialization, (ii) refinement stage implementing a swapping protocol that exchanges mask values between each pair of masked and unmasked parameters within the same layer, and (iii) tuning stage to relax the binary mask values into continuous values using a reconstruction objective. To validate the necessity of refinement stage, we compare the model performance with and without the refinement stage. We first fine-tune a full-layer LLaMA2-7B model with Databricks-Dolly-15K, and using FLoE to get the layer importance rankings which is shown in Figure 9. Empirical results reveal that disabling this stage leads to a performance degradation as shown in Table 7.

Table 7: Evaluation results on MMLU with the former 15, 16, 21, 24, 25 layers fine-tuned (in the ranking order).

| Fine-tuned Layers | 15 | 16 | 21 | 24 | 25 |
|---|---|---|---|---|---|
| w/o refinement | 45.11 | 45.14 | 45.52 | **45.48** | 45.61 |
| w/ refinement | **45.36**$^{\uparrow 0.25}$ | **45.33**$^{\uparrow 0.19}$ | **45.56**$^{\uparrow 0.04}$ | 45.43$^{\downarrow 0.05}$ | **45.66**$^{\uparrow 0.05}$ |

## G  ANALYSIS FOR LAYER SELECTION

As FLoE layer selection algorithm need to run on an already fine-tuned model, we decrease the preparation overhead via reducing the size and category of training data. For single domain fine-

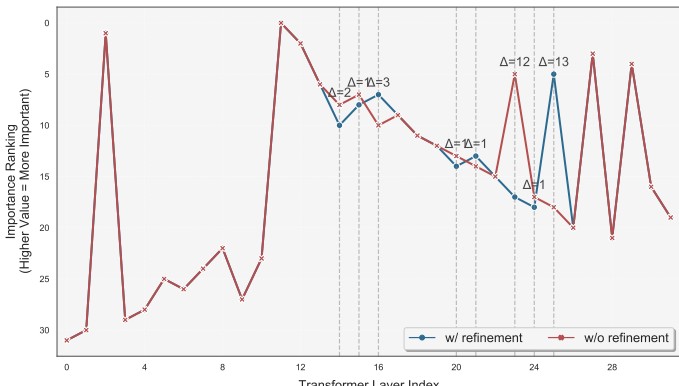

Figure 6: Layer importance rankings with or without refinement stage. The difference starts at the rank 14 (i.e. the former 15 layers. The 14-th layer is layer 10 in the w/o refinement rankings and layer 8 in the w/ refinement rankings) and ends at the rank 16 (i.e. the former 17 layers has the same set of layer indexes). Similarly, we only need to evaluate the former 21, 24 and 25 layers in the subsequent rankings. Both rankings and layer indexes start at 0. The range is 0-31.

tuning, we first fine-tune a model on a general knowledge dataset, then run FLoE on that model. If this general dataset is too large, we reduce its size by randomly selecting a partial of the training data. Note that we can use the selection results for other single domain fine-tuning including medical, law, code and mathematics.

**Different Size of Sample Dataset.** To investigate the impact of training data quantity on layer selection, we conduct a comparative study using MoE-based LoRA for LLaMA2-7B fine-tuning. We evaluate layer importance rankings across three training scenarios: full-data (100%), moderately reduced (70%), and substantially reduced (30%) subsets of the Databricks-Dolly-15K dataset.

Figure 7 illustrates the stability of layer rankings under these varying data regimes. Our analysis reveals that the refinement stage exhibits minimal sensitivity to training data quantity in terms of rank consistency. Across all three configurations, layers 0-2 and layers 10-14 demonstrate strong agreement in stability rankings. However, notable discrepancies emerge in the final layers (22-28), suggesting that deeper layers show higher variability in importance assessment when trained with reduced data. This observation indicates that for a safe consideration, we should choose at least 50% layers for subsequent fine-tuning on the target dataset. For a more careful layer selection, we should enlarge the sample dataset size.

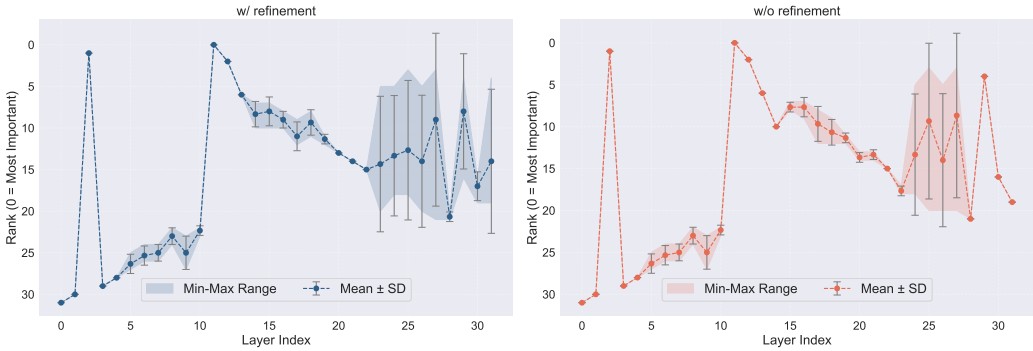

Figure 7: Rank stability comparison of layers with (left) and without (right) refinement stage. The plots show layer importance rankings (lower rank indicate higher importance) across three training data ratios (100%, 70%, 30%). The shaded regions depict min-max ranges of the rank on different ratios.

**Fine-tune on Different Target Datasets.** After ranking layers on the sample dataset, we apply the ranking results to selectively fine-tune the base model on the target datasets. Note that the

target dataset can be different from the sample dataset. To evaluate the **generalization** capability of FLoE, we compare its fine-tuning performance with two single domain datasets (Clinic-10K and CodeAlpaca). For each dataset, we implement two layer selection strategies: one using the general knowledge dataset (Databricks-Dolly-15K) as the sample dataset, and the other using the target dataset itself as the sample dataset. Figure 8 shows that although the misalignment between the sample and target datasets has little impact on the best achievable performance, it may affect the number of layers required to reach that performance.

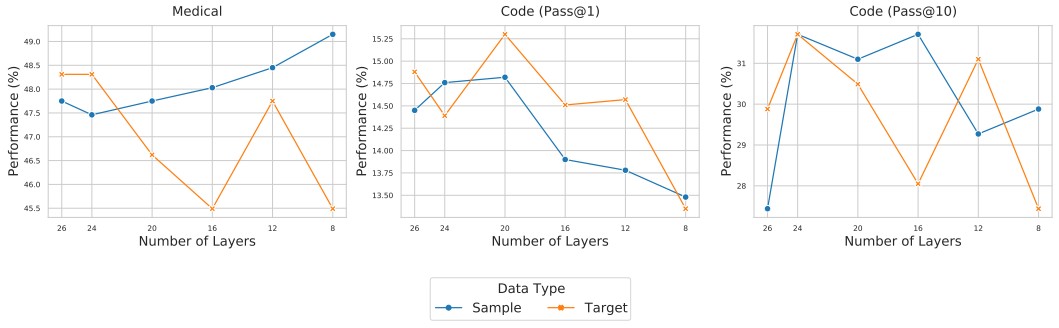

Figure 8: Illustration of performance under different layer importance rankings.

## H   PROOF OF EQ. 16

The tuning stage is designed to minimize the layer-wise reconstruction error between the masked and unmasked versions of the adapted model. The goal is to optimize $\mathbf{m}_k$ such that the squared $L_2$-norm of the residual activation difference is minimized:

$$\underset{\tilde{\mathbf{m}}_k}{\arg\min} \|x' + l_k(x'; \mathbf{m}_k) - (x + l_k(x; \mathbb{1}))\|_2^2 \quad \text{s.t.} \quad Z(\mathbf{m}_k) = Z(\bar{\mathbf{m}}_k). \tag{17}$$

where $Z(\mathbf{m})$ denotes the indices of zero entries in $\mathbf{m}$. We use $\bar{\mathbf{m}}$ to indicate the fixed binary mask values obtained from the refinement stage, and $\mathbf{m}$ indicates the mask variables to be optimized, reconstructed from the binary masks. Let $\mathbf{W}_i$ represent the $i$-th head or neuron weight of layer $l_k$. The output of $l_k(\cdot; \mathbf{m}_k)$ is:

$$l_k(x; \mathbf{m}_k) = \sum_{i=1}^{N} \mathbf{m}_{k,i} \mathbf{W}_i x, \tag{18}$$

where $N$ denote the number of heads (for MHA) or neurons (for FFN). Let $\Delta x = x' - x$, then the residual difference in Eq. 17 becomes:

$$x' + \sum_{i=1}^{N} \mathbf{m}_{k,i} \mathbf{W}_i x' - x - \sum_{i=1}^{N} \mathbb{1}_i \mathbf{W}_i x = \sum_{i=1}^{N} \mathbf{m}_{k,i} \mathbf{W}_i (x + \Delta x) + \Delta x - \sum_{i=1}^{N} \mathbb{1}_i \mathbf{W}_i x \tag{19}$$

$$= \sum_{i=1}^{N} \bar{\mathbf{m}}_{k,i} \mathbf{m}_{k,i} \mathbf{W}_i (x + \Delta x) + \Delta x - \sum_{i=1}^{N} \mathbb{1}_i \mathbf{W}_i x$$

Let $\mathbf{A} = \sum_{i=1}^{N} \bar{\mathbf{m}}_{k,i} \mathbf{W}_i (x + \Delta x)$, $\mathbf{b} = \Delta x - \sum_{i=1}^{N} \mathbb{1}_i \mathbf{W}_i x$. Then Eq. 17 can be transformed into a linear least-square problem:

$$\underset{\mathbf{u}_k}{\arg\min} \|\mathbf{A}\mathbf{m}_k - \mathbf{b}\|_2^2. \tag{20}$$

Assume $\mathbf{A}^\top \mathbf{A}$ is invertible. The closed-form solution to Eq. 20 is given by:

$$\mathbf{m}_k^* = (\mathbf{A}^\top \mathbf{A})^{-1} \mathbf{A}^\top \mathbf{b}. \tag{21}$$

## I   MORE RESULTS

---

**Algorithm 2** Joint Mask Refinement

---

1: **Input:** Initial masks $\mathcal{M}_h^*, \mathcal{M}_f^*$, budget $C$, Fisher/Taylor scores.
2: Compute initial loss $\mathcal{L}^*$ and used budget $C_{\text{used}} = C - \left( \sum_{i \notin \mathcal{M}_h^*} t_h + \sum_{j \notin \mathcal{M}_f^*} t_f \right)$
3: **while** improvement **do**
4:      Generate all valid swaps $(p,q)$ where $\Delta C_{p \to q} \geq 0$
5:      Select swap $(p^*, q^*) = \arg\max_{(p,q)} \Delta\mathcal{L}_{p \to q}$
6:      **if** $\Delta\mathcal{L}_{p^* \to q^*} > 0$ **then**
7:          Update $\mathcal{M}_h^*, \mathcal{M}_f^*, \mathcal{L}^* \leftarrow \mathcal{L}^* - \Delta\mathcal{L}_{p^* \to q^*}$
8:          Update $C_{\text{used}} \leftarrow C_{\text{used}} + \Delta C_{p^* \to q^*}$
9:      **else**
10:         **break**
11:      **end if**
12: **end while**
13: **Output:** Refined masks $\mathcal{M}_h^*, \mathcal{M}_f^*$

---

Table 8: Search efficiency of different hyperparameter tuning strategies on $r \in [4, 64]$ (with step of 4) and number of experts $\in [2, 10]$.

| Method | Search Space Size | Trials Required | Relative Cost |
|---|---|---|---|
| Grid Search | $16 \times 9 = 144$ | 144 | $100\%$ |
| Random Search | $16 \times 9 = 144$ | $\sim 80$ | $\sim 55\%$ |
| Bayesian Opt. | $16 \times 9 = 144$ | $\sim 30$ | $\sim 21\%$ |

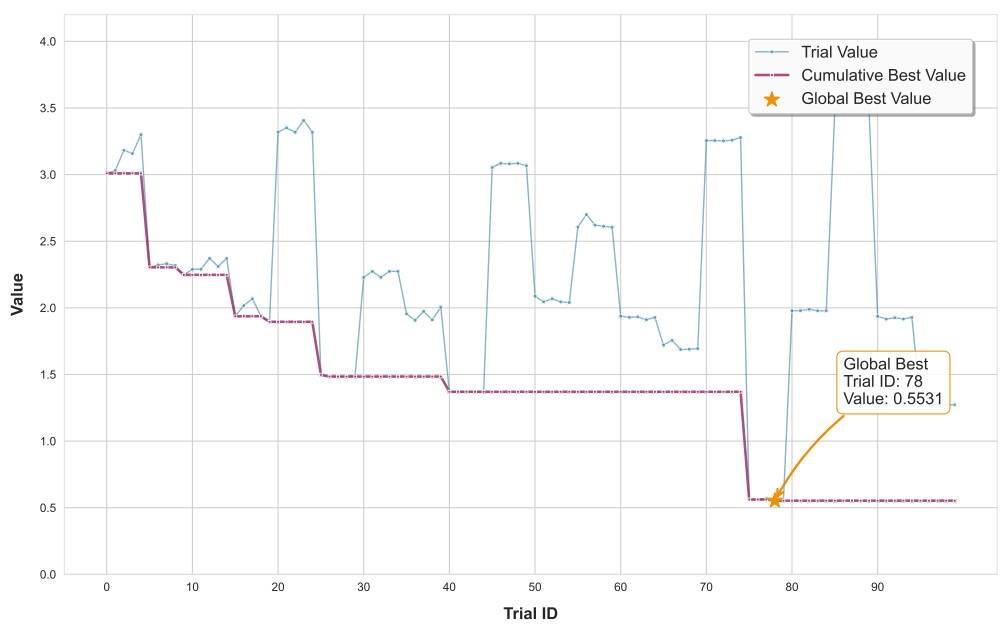

Figure 9: Bayesian optimization process for LLaMA3.2-3b. The validation loss converges between 30–70 trials. Although the global best value achieved at trial 78, convergence is already evident by trial 40.

Table 9: Detailed results of BBH evaluation with LLaMA2-7B as the base LLM (3-shot).

| Task | Base | LoRA | HydraLoRA L=32 | Ours (FLoE) L=28 | L=24 | L=20 | L=16 | L=12 | L=8 |
|---|---|---|---|---|---|---|---|---|---|
| Boolean Expressions | 73.60 | 74.40 | 75.20 | 77.20 | 76.01 | 76.40 | **77.60** | 74.07 | 71.35 |
| Causal Judgement | 47.06 | 54.90 | 57.22 | 56.15 | 50.72 | 57.14 | 50.27 | 55.08 | **59.89** |
| Date Understanding | 37.60 | 40.00 | 44.80 | 55.60 | 54.77 | 56.40 | 57.20 | 59.60 | **60.80** |
| Disambiguation QA | 34.80 | 45.20 | 56.80 | 60.00 | 65.60 | 64.00 | 64.40 | 53.17 | **66.00** |
| Dyck Languages | 10.80 | 13.20 | 14.40 | **15.60** | 13.20 | 14.06 | 14.92 | 14.49 | 14.00 |
| Formal Fallacies | 44.80 | 46.00 | 46.80 | 46.40 | 46.80 | 47.60 | 48.00 | 47.15 | **49.20** |
| Geometric Shapes | 9.70 | 10.20 | 15.20 | 14.40 | **25.60** | 19.60 | 12.40 | 12.77 | 12.00 |
| Hyperbaton | 30.80 | 40.00 | 48.40 | 48.80 | 48.40 | 52.00 | **53.20** | 48.40 | 49.20 |
| Logical Deduction (five objects) | 22.80 | 33.30 | 45.60 | **46.80** | 45.13 | 45.53 | 46.34 | 43.10 | 46.00 |
| Logical Deduction (seven objects) | 16.00 | 22.40 | 32.00 | 31.60 | 30.80 | **32.40** | 30.40 | 30.00 | 29.60 |
| Logical Deduction (three objects) | 35.20 | 41.40 | 44.40 | 68.40 | 59.60 | 61.20 | 52.00 | 56.40 | **69.60** |
| Movie Recommendation | 53.50 | 63.05 | 68.67 | **69.88** | 66.51 | 69.08 | 68.27 | 65.72 | 65.72 |
| Multistep Arithmetic | 0.80 | 0.80 | 1.20 | **1.60** | 1.36 | 1.26 | 1.26 | 1.26 | 1.20 |
| Navigate | 42.40 | 52.70 | 57.10 | 54.40 | 54.40 | 62.80 | 62.80 | 58.40 | **65.20** |
| Object Counting | 40.10 | 44.00 | 42.40 | 44.00 | 45.20 | 46.00 | 44.80 | 46.00 | **50.00** |
| Penguins in a Table | 21.70 | 22.60 | 26.03 | 28.08 | 33.56 | 28.08 | 28.77 | 33.56 | **48.63** |
| Reasoning about Colored Objects | 19.40 | 27.20 | 35.60 | 40.00 | 36.59 | 42.00 | 41.60 | **42.40** | 37.60 |
| Ruin Names | 25.40 | 28.70 | 30.65 | 33.47 | 31.92 | **37.90** | 34.68 | 34.27 | 29.03 |
| Salient Translation Error Detection | 11.20 | 25.20 | 26.80 | 26.00 | 27.95 | **32.40** | 32.00 | 32.00 | 30.40 |
| Snarks | 44.00 | 44.00 | 46.63 | **48.31** | 46.63 | 46.63 | 47.19 | 46.63 | 46.63 |
| Sports Understanding | 50.00 | 57.20 | 65.60 | **67.20** | 58.00 | 57.60 | 54.80 | 55.20 | 58.00 |
| Temporal Sequences | 21.10 | 32.60 | 33.20 | 33.60 | 34.80 | 32.90 | 32.90 | 33.60 | **36.40** |
| Tracking Shuffled Objects (five objects) | 21.90 | 31.20 | 37.20 | 37.60 | 38.40 | **40.00** | 38.80 | 38.40 | 39.20 |
| Tracking Shuffled Objects (seven objects) | 14.80 | 14.00 | 26.00 | 27.60 | **29.60** | 26.40 | 28.80 | 28.40 | 27.60 |
| Tracking Shuffled Objects (three objects) | 41.20 | 51.20 | 71.20 | 71.60 | **72.80** | 70.00 | 69.20 | 67.60 | 65.60 |
| Web of Lies | 48.80 | 51.20 | 50.80 | 50.00 | 51.97 | 52.80 | 51.20 | **53.20** | 50.40 |
| Word Sorting | 22.00 | 21.20 | 22.40 | 20.40 | 22.00 | 22.40 | **24.40** | 23.60 | 22.00 |
| Avg Performance (EM) | 31.17 | 36.59 | 41.57 | 43.51 | 42.73 | 44.24 | 43.27 | 42.76 | **44.49** |
| # of A/B for training/inference | 0/0 | 1/1 | 1/6 | 1/6 | 1/6 | 1/6 | 1/6 | 1/6 | 1/6 |
| % Params | - | 0.062 | 0.205 | 0.179 | 0.153 | 0.128 | 0.102 | 0.077 | 0.051 |

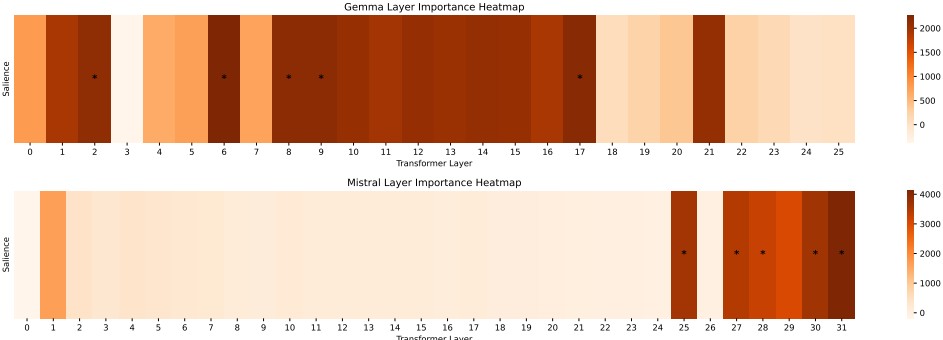

Figure 10: Layer importance heatmaps for Gemma2-2B (top) and Mistral-7B (bottom), highlighting critical adaptation layers (8-17 for Gemma2-2B; 27-31 for Mistral-7B). Saliency values reflect the contribution of each layer, with darker hues indicating higher importance.

