# OpenReview forum: "FLoE: Fisher-Based Layer Selection for Efficient Sparse Adaptation of Low-Rank Experts"
_ICLR.cc/2026/Conference — ICLR 2026 Conference Withdrawn Submission_

### Official Review · Reviewer_d6zf · 2025-10-25

**Soundness:** 2
**Presentation:** 2
**Contribution:** 2
**Rating:** 4
**Confidence:** 4

**Summary:**

This paper introduces FLoE, a parameter-efficient fine-tuning (PEFT) framework for large language models. It proposes a Fisher information-guided scoring to select a sparse, task-critical subset of transformer layers for low-rank expert adaptation, and a Bayesian optimization procedure for automatically allocating LoRA ranks without costly grid searches. FLoE is empirically validated on multiple LLMs and domains, claiming to match or exceed competitive PEFT baselines with substantial parameter and computational savings.

**Strengths:**

- **Principled Layer Selection**: FLoE’s utilization of Fisher information to guide which layers receive adapters is theoretically motivated and systematically implemented. This is a marked improvement over uniform or brute-force selection.
- **Dynamic Rank Allocation**: The inclusion of Bayesian optimization for LoRA rank and expert allocation addresses a significant practical bottleneck: the need for extensive hyperparameter tuning.
- **Comprehensive Experiments**: Evaluations span multiple settings, several LLM families (LLaMA2-7b, Mistral-7b, Gemma2-2b, and LLaMA3.1-8b), and diverse tasks (MMLU, GSM8K, HumanEval, BBH).

**Weaknesses:**

- Absence of “Ablation” for Fisher vs. Other Importance Measures: The methodology justifies Fisher information–based selection, but there is no systematic comparison against alternative (e.g., gradient norm, attention-based, or purely empirical) layer-importance metrics. The ablation in Table 7 focuses on the mask refinement step rather than the core Fisher metric. This omission prevents a full understanding of why FLoE’s particular mechanism is essential. Even simpler ablations methods like the prioritizing the opposite of the layers that FLoE deemed important are missing.
- The limitations section mentions that future work can explore dynamic expert allocation mechanisms, where both the selection of critical layers and the number of experts per layer are jointly optimized. But several papers already explore that which were not compared against. One of them is MoLA which this paper cites, but other ones such as AlphaLora [1] and LayerIF [2] which the paper doesn't cite or compare against.
- (Minor Weakness) Over reliance on MoE Derivative Architectures: The method is evaluated nearly exclusively in the context of MoE LoRA or its asymmetric variants. It is not studied if the Fisher-based and Bayesian strategies transfer outside of this architecture family (e.g., plain Adapter-based or Prompt-based PEFT methods). Thus, the generalizability of FLoE is not validated.
- The paper mentioned that there will be a code release in the supplementary but a link that was not provided.

---
References:

[1]. [AlphaLoRA: Assigning LoRA Experts Based on Layer Training Quality](https://aclanthology.org/2024.emnlp-main.1141/) (Qing et al., EMNLP 2024)
[2]. Askari, Hadi, et al. "LayerIF: Estimating Layer Quality for Large Language Models using Influence Functions." Neurips 2025.

**Questions:**

- How does the computation and memory overhead of the layer selection (warm-up) process scale with model size and number of tasks? Are there ways this overhead can be amortized or further reduced for frequent new-task adaptation?
- Why were alternative layer importance metrics (beyond Fisher and Taylor) not compared in ablation?
If such ablations exist, pointing to them could bolster confidence in FLoE’s approach.
- Have the authors run statistical significance tests (e.g., t-test or bootstrapping) to support the claimed performance gains? If not, can they report results over repeated runs?

---

### Official Review · Reviewer_UTou · 2025-10-26

**Soundness:** 2
**Presentation:** 3
**Contribution:** 2
**Rating:** 6
**Confidence:** 3

**Summary:**

The paper presents FLoE to fix a key problem with current LoRA methods: they add adapters to every MoE layer.
FLoE scores layers using Fisher information to adapt only the important ones, and uses Bayesian optimization to pick the right LoRA rank for each dataset without brute-force tuning.
Tests across several language models and domains show FLoE keeps performance competitive while cutting memory, training cost, and inference latency.

**Strengths:**

- FLoE tackles both redundancy and inefficiency in current PEFT approaches.
- The study provides extensive baseline comparisons and detailed latency/runtime studies.

**Weaknesses:**

- The integration of layer selection and adaptive rank tuning is nice, but each component has been explored on its own in prior studies.
- The method leans on a proxy dataset, but there is no ablation on the proxy.
- The layer-importance scoring hinges on Fisher information computed from pretraining loss sensitivity, implicitly assuming access to reliable pretraining gradients.

**Questions:**

see Weaknesses

---

### Official Review · Reviewer_qtGT · 2025-10-31

**Soundness:** 3
**Presentation:** 3
**Contribution:** 2
**Rating:** 4
**Confidence:** 4

**Summary:**

This paper introduces FLoE, a novel PEFT method designed to address the inefficiencies of standard LoRA deployment. The authors identify that uniformly applying adapters across all layers is redundant and suboptimal. FLoE proposes a two-part solution: A Fisher information-guided scoring mechanism to identify and sparsely deploy adapters only on task-critical Transformer layers, and A Bayesian optimization-driven allocator to automatically and efficiently determine the optimal LoRA rank.

**Strengths:**

- The experimental setting of this paper is solid, and its performance surpasses several baselines.

- This paper provides an automated and theoretically-grounded solution for layer selection, addressing a practical challenge in PEFT.

**Weaknesses:**

- The method has limitations as it primarily focuses on the LoRA-MoE scenario. While this appears fancy, it is not the primary application context for standard LoRA.

- A comparison with relevant work is lacking. Many studies are optimizing LoRA from an efficiency perspective, with some focusing on rank reduction (e.g., AdaLoRA, which was included) and others on parameter sharing. The baselines compared in this paper could be more comprehensive to better situate the work. Some related works:

https://arxiv.org/html/2410.11772v1

https://arxiv.org/html/2402.08562v1

https://openreview.net/pdf?id=Thv66GmqZS

https://arxiv.org/abs/2406.10785

https://openreview.net/pdf?id=O6QZ4W6GXt

https://arxiv.org/abs/2410.19694

- The proposed method is not particularly novel and is quite complex. The "less is more" concept in the PEFT field (i.e., sparse adaptation) has been explored by many other works.

**Questions:**

- It is unclear whether the performance degradation over training is due to overfitting. The paper would be strengthened by providing a comparative analysis of performance at different training stages against other baselines to clarify this point.

- It remains unclear how much advantage the proposed method holds on more powerful, state-of-the-art models, such as Qwen2.5 or Qwen3.

---

### Official Review · Reviewer_gr73 · 2025-11-01

**Soundness:** 2
**Presentation:** 2
**Contribution:** 2
**Rating:** 4
**Confidence:** 3

**Summary:**

The paper proposes an algorithm for fine-tuning model using the MoE variant of LoRA, tackling the problem of finding the optimal ranks for the adapters. The overview of the method is as follows. First, the method uses a small training dataset to warm up MoE LoRA (same variant proposed by HydraLoRA). This part employs a Bayes optimization Then the model uses an algorithm to determine a masking of the base model to see which are the important weights to be tuned. This is done by viewing the optimization problem for the mask a constrained optimization problem with Fisher information matrix and the constraints involve the cost determined by Taylor importance. Finally the model is tuned on the larger dataset. Experiments show that the proposed method performs well on several tasks with improved runtime compared with full fine-tuning. Additional insights are provided for the layer importance.

**Strengths:**

- Finding the layer importance and the optimal ranks for the adapters is a challenging problem and this paper provides a new viewpoint using Bayesian optimization and Fisher information to solve this problem.
 - The experiments are quite comprehensive and show pretty good results.

**Weaknesses:**

- First, for the Bayesian optimization step, the paper missed out the following paper which also uses Bayesian optimization for finding the optimal ranks in LoRA (though not MoE + LoRA)
[1] AutoPEFT: Automatic Configuration Search for Parameter-Efficient Fine-Tuning

- For optimizing the mask:
  - The paper doesn’t seem to consider the problem that the mask requires the same number of parameters as the model (unless I’m missing something here). So what confuses me here is what makes this part more memory-efficient compared with full fine-tune?

- There are several questions in this part
  - In Line 196, regarding the first order term. Here the gradient is with respect to m, so it is not 0
  - approximating the hessian with the outer product of the gradients is general not true. In the case of Fisher information (which typically is used for the log likelihood), the hessian has the opposite sign to the outer product of the gradients. So the justification for this step is not valid.
  ⁃ There are several place where typos make it very confusing. Line 161-162 the expressions should read $Z(m_k)$ and $Z(1-m_l)$ / or $Z(\bar{m}_l)$. In Eq. 12, the constraints should be the sum over $i \in (\bar{m}_l)$. Otherwise, the problem has a trivial solution of $Z(m_k) = \emptyset$.

- The paper can be improved in terms of the clarity and presentation. In particular, I don’t understand what is done after solving the constraint problem. How are the ranks of adapters determined?

**Questions:**

- See weaknesses
- Fisher information is often used with the log likelihood. Why is the product of the gradients in Eq. 8 called Fisher information?
- Please also report the memory requirement in the experiment.

---

### Note · Authors · 2026-01-13

I have read and agree with the venue's withdrawal policy on behalf of myself and my co-authors.